



# Active heat pulse sensing of 3D-flow fields in streambeds

Eddie W. Banks* [1], Margaret A. Shanafield [1], Saskia Noorduijn [1], James McCallum [1], Jörg Lewandowski [2,3], Okke Batelaan [1]

[1] National Centre for Groundwater Research and Training and the College of Science and Engineering, Flinders University,

Adelaide, South Australia, Australia.

[2] IGB, Leibniz-Institute of Freshwater Ecology and Inland Fisheries, Department Ecohydrology, Berlin, Germany.

[3] Humboldt University Berlin, Geography Department, Berlin, Germany.

*Correspondence to*: Eddie W Banks (eddie.banks@flinders.edu.au)

**Abstract.** Profiles of temperature time series are commonly used to determine hyporheic flow patterns and hydraulic dynamics

in the streambed sediments. Although hyporheic flows are 3D, past research has focused on determining the magnitude of the vertical flow component and how this varies spatially. This study used a portable 56 sensor, 3D temperature array with 3 heat pulse sources to measure the flow direction and magnitude up to 200 mm below the water-sediment interface. Short, one-minute heat pulses were injected at one of the three heat sources and the temperature response was monitored over a period of 30 minutes. Breakthrough curves from each of the sensors were analyzed using a heat transport equation. Parameter estimation

and uncertainty analysis was undertaken using the DREAM algorithm, an adaption of the Markov chain Monte Carlo method, to estimate the flux and its orientation. Measurements were conducted in the field and in a sand tank under an extensive range of controlled hydraulic conditions to validate the method. The use of short duration heat pulses provided a rapid, accurate assessment technique for determining dynamic and multi-directional flow patterns in the hyporheic zone and is a basis for improved understanding of biogeochemical processes at the water-streambed interface.

## 1 Introduction

Application of heat as a tracer to hydrological studies has rapidly progressed in recent decades, driven by the simplicity of the methodology and low cost of sensor technology (Anderson, 2005; Rau et al., 2014). Using this method, spatial and temporal flow dynamics within the hyporheic zone, particularly hyporheic transport and exchange (e.g. longer attenuation), have been shown to enhance stream denitrification (Harvey et al., 2013; Gomez-Velez et al., 2015; Zarnetske et al., 2011), degradation of

mine-pollutants (Gandy et al., 2007) and the degradation of wastewater micro-pollutants (Engelhardt et al., 2013). It is also widely used by other disciplines e.g. ecology where the thermal regime in river systems plays an important role in ecosystem health (Caissie, 2006; Harvey and Wagner, 2000; Brunke and Gonser, 1997; Boulton et al., 1998).

The majority of streambed heat tracer studies use vertical, ambient temperature profiles and a one-dimensional analytical solution of the heat diffusion-advection equation to estimate streambed exchange fluxes and infer hyporheic flow patterns

(Constantz et al., 2002; Naranjo and Turcotte, 2015; Rau et al., 2010; Vogt et al., 2010). When using ambient temperature





fluctuations and one dimensional heat transport models, several days of data are required to estimate the vertical flux. In addition, an assumption is made that the dominant exchange process is in the vertical direction only and the horizontal or lateral component of flow is considered to be negligible. There are very few investigations which have tried to capture both the vertical and horizontal component of flow, as the determination of the non-vertical component is challenging (Munz et al.,

2016;Briggs et al., 2012;Shanafield et al., 2016).

More recently, the suitability of active temperature sensing has been explored as an approach to characterise streambed spatial and temporal exchange dynamics in three dimensions. The injection of heat as a tracer is not new, with a number of studies using the active temperature sensing technique to evaluate groundwater flow within wells (Sellwood et al., 2016;Read et al., 2014;Banks et al., 2014), flow in sediments (Greswell et al., 2009;Ballard, 1996;Bakker et al., 2015), surface water-

groundwater exchange processes (Kurth et al., 2015) and hyporheic exchange flows in the hyporheic zone (Angermann et al., 2012a;Angermann et al., 2012b;Lewandowski et al., 2011).

The aim of the present study was to develop an active heat pulse sensing (HPS) instrument to conduct rapid assessments of the three-dimensional (3D) flow field in the streambed from fine silt to coarse gravels across different geomorphological structures. It builds upon previous studies by Lewandowski et al. (2011) and Angermann (2012a;2012b), who developed an

active heat pulse sensor to determine the flow direction and flow velocity in shallow sandy stream environments. In the present study we have developed a more robust field instrument and advanced the analysis of the temperature breakthrough data using the analytical solution of the heat transport equation for a 3D array. Parameter estimation and uncertainty analysis was implemented using the DREAM algorithm (Vrugt et al., 2009c), an adaption of the standard Markov chain Monte Carlo, to determine the direction and magnitude of flow velocity patterns in the streambed at multiple depths. Laboratory tests were

conducted in a sand tank using an extensive range of flow scenarios with tightly controlled hydraulic conditions to evaluate the methodology. Field tests demonstrated the active heat pulse instrument in different geomorphological structures in a small stream in the Mount Lofty Ranges, South Australia.

## 2 Material and Methods

### 2.1 General Design and Operating Principles

A 56 sensor, 3D temperature array with 3 heat pulse sources (a.k.a. the Hot Rod) was developed to measure the flow direction and magnitude up to 200 mm below the water-sediment interface in the streambed (Figure 1). The central carbon-fibre rod (260 mm long with a diameter of 12 mm) has three equally spaced, 60 watt heating elements along its length at positions of 65, 140 and 215 mm below the base plate and are referred to as heat injection depth R1, R2 and R3, respectively (Figure 1). Eight stainless steel rods (6 mm diameter and 298 mm long) housing 7 equally spaced (38 mm apart) temperature thermistors

(Maxim DS18B20; precision 0.06 degrees) are arranged cylindrically around the carbon-fibre rod and at two fixed spacings of 28 and 47 mm (Table S1. SI). The central rod and thermistor sticks are fixed to a rigid circular aluminium base plate which is attached to a collapsible handle. The material, dimensions and spacing of the rods to the base plate was designed to reduce



flexibility and minimise disturbance to the sediment material on insertion, as this was a limitation in previous studies. A critical design feature was to ensure that the rods stayed parallel to one another and that the spacing did not vary during installation, as it was designed to be used in a range of sedimentary environments from fine silt to coarse gravels.

A terminal program is used to communicate with the data logger and to control the sampling routine of the Hot Rod (e.g. 5 sampling frequency, duration and power output of the active heat pulse, selected heating element used and logging period). A sealed 12 volt lead acid battery together with a power supply regulator is used to maintain a constant 12 volt output to the heating elements. The power delivered to the three heating elements can be adjusted in the logger program from 0 to 100 % to provide greater flexibility to the required active heat pulse. The input current from the power supply regulator is also recorded each time the temperature is measured to ensure a tight control on the actual power being delivered through the heating 10 elements to the surrounding material and therefore reducing uncertainty in the analysis routine.

An important aspect of the design was that it could be rapidly and easily deployed to capture large spatial data sets along a reach of stream or across a pool-riffle sequence. Installation requires gently pushing (or lightly tapping with a shockless impact hammer) the device into the streambed ensuring that there is a sufficient gap between the top of the sediment and the underside of the base plate to prevent streamflow constriction (the top thermistor is set at 44 mm below the baseplate so a gap of ~30 15 mm puts the first thermistor just below the sediment/water interface). The impact of the installed device on flow velocity and direction is expected to be minimal given that the volume of the device relative to the volume of the measurement area is less than 4%.

Once installed and equilibrated with the surrounding sediment, the logger program is executed and the ambient temperature is measured at each thermistor ($T_0$), directly followed by activation of the selected heat element for the chosen duration. The data 20 logger records the temperature differential ($\Delta T = T_t - T_0$) at the chosen sample frequency to clearly discern the timing and location of the breakthrough curve at each of the thermistors. Field and laboratory tests showed that short, 1-min heat pulse injections and a 20 – 30 min temperature response monitoring period is appropriate for estimating the dominant direction and flux magnitude in sandy streambeds. Specific details of the analysis routine that was used are described in the following section.

## 25 2.2 Data Analysis and Routine Outputs

### 2.2.1 Heat Transport Simulation

The magnitude and direction of the pore water velocity at the observation point based on the measured temperature breakthrough curves at the 56 sensors was determined using a modified version of the heat transport equation:

$$\frac{\partial T}{\partial t} = \nabla \left( \frac{\kappa_0}{\rho c} \nabla T \right) - \nabla \left( \frac{\rho_w c_w}{\rho c} qT \right)$$

(1)





where $T$ is temperature (ºC), $\kappa_0$ is the bulk thermal conductivity of the water saturated sediments (W m$^{-1}$ºC$^{-1}$), $\rho c$ is the volumetric heat capacity of the water saturated sediments (J m$^{-3}$ ºC$^{-1}$), $\rho_w c_w$ is the volumetric heat capacity of water (4.1 J m$^{-3}$ ºC$^{-1}$) and $q$ is the Darcy velocity (or Darcy flux) of water (m s$^{-1}$).

An analogy can be made to the solute transport equation where the mean pore water velocity is replaced with $\dfrac{\rho_w c_w}{\rho c} q$ and the

5  dispersion tensor can be replaced with $\mathbf{I}\left(\dfrac{\kappa_0}{\rho c}\right)$, where $\mathbf{I}$ is the identity matrix. Assuming that these components are constant

in space and time, Equation 1 can be reduced to:

$$\frac{\partial T}{\partial t} = \frac{\kappa_0}{\rho c} \nabla^2 T - \frac{\rho_w c_w}{\rho c} q \nabla T$$

(2)

For an instantaneous injection of temperature in an infinite, one-dimensional aquifer, a solution is obtained by modifying the

10  solution of Bear (1972):

$$T(x,t) = \frac{T_0}{\sqrt{4\pi \frac{\kappa_0}{\rho c} t}} \exp\left(-\frac{\left(x - \frac{\rho_w c_w}{\rho c} qt\right)^2}{4\frac{\kappa_0}{\rho c} t}\right)$$

(3)

Three–dimensional solutions also exist where there is only an $x$ component of velocities (Domenico and Schwartz, 1998):

$$T(x, y, z, t) = \frac{M_0}{\rho c \left(4\pi \frac{\kappa_0}{\rho c} t\right)^{\frac{3}{2}}} \exp\left(-\frac{\left(x - \frac{\rho_w c_w}{\rho c} qt\right)^2}{4\frac{\kappa_0}{\rho c} t} - \frac{y^2}{4\frac{\kappa_0}{\rho c} t} - \frac{z^2}{4\frac{\kappa_0}{\rho c} t}\right)$$

(4)

where $M_0$ is the thermal mass (W).

The thermal mass input was not considered in previous studies as a known parameter (Angermann et al., 2012b). The Hot

20  Rod, however, uses a known wattage, and the thermal mass term is given as:

$$M_0 = F \cdot dt$$

(5)





where $F$ is the heat flux and $dt$ is the duration of application. The thermal mass input, $M_0$ is measured by the data logger such that at full power, the theoretical output of the 60 watt heating element provides 5 amps of current and an injection period of 60 seconds equals an energy input of 3600 joules (J).

It is not always the case that the fluxes will be uni-directional and therefore the location of the observations can be converted through rotation of the coordinate system so that flow is in the x direction. In this application we assume that the vector of Darcy fluxes in the $x$, $y$ and $z$ direction is defined as:

$$\mathbf{q} = \begin{bmatrix} q_x \\ q_y \\ q_z \end{bmatrix}$$

(6)

The coordinate system is first rotated such that the points are orientated around the z axis and we define the angle $\theta$ (Figure 2), where:

$$\theta = \tan^{-1}\left(\frac{q_y}{q_x}\right)$$

(7)

The rotational Jacobian matrix is then defined as:

$$\mathbf{J} = \begin{bmatrix} \cos\theta & \sin\theta & 0 \\ -\sin\theta & \cos\theta & 0 \\ 0 & 0 & 1 \end{bmatrix}$$

(8)

And the coordinates:

$$\mathbf{x}' = \mathbf{J}\mathbf{x}$$

(9)

And the flux:

$$\mathbf{q}' = \mathbf{J}\mathbf{q}$$

(10)

Secondly, we rotate the points around the y axis. To do this we define the angle $\phi$, where:

$$\phi = \tan^{-1}\left(\frac{q'_z}{q'_x}\right)$$

(11)

The rotational Jacobian matrix is then defined as:

$$\mathbf{J} = \begin{bmatrix} \cos\phi & 0 & \sin\phi \\ 0 & 1 & 0 \\ -\sin\phi & 0 & \cos\phi \end{bmatrix}$$

(12)

And the coordinates:





$$\mathbf{x''} = \mathbf{Jx'} \tag{13}$$

And the flux:

$$\mathbf{q''} = \mathbf{Jq'} \tag{14}$$

The transformation results in the representation of the Darcy fluxes as $q''_x = \|\mathbf{q}\|$ (the magnitude of the non-transformed flow

vector) and $q''_y = q''_z = 0$. The distances of the sensors are oriented relative to this new flow vector. The main advantage

over previous approaches is that all sensors can be included rather than a single sensor from the array accounting for a single

transverse distance (Lewandowski et al., 2011).

We then substitute these new dimensions into Equation 4 to get:

$$T(x'', y'', z'', t) = \frac{M_0}{\rho c \left(4\pi \frac{\kappa_0}{\rho c} t\right)^{\frac{3}{2}}} \exp\left(-\frac{\left(x'' - \frac{\rho_w c_w}{\rho c}\|\mathbf{q}\|t\right)^2}{4\frac{\kappa_0}{\rho c} t} - \frac{y''^2}{4\frac{\kappa_0}{\rho c} t} - \frac{z''^2}{4\frac{\kappa_0}{\rho c} t}\right) \tag{15}$$

To account for the variation in the boundary condition, Equation 15 was implemented for a series of discrete pulses. The use
of discrete pulses is implemented as:

$$T(x'', y'', z'', t) = \sum_{\tau=t_{on}}^{t_{off}} T(x'', y'', z'', t - \tau) \tag{16}$$

where $t_{on}$ is the time at which the heating element was turned on and $t_{off}$ is the time when the heating element was turned off.

The operator $\tau$ represents the time lags, and for each discrete time, $M_o = F \cdot d\tau$. This method allows for the

representation of a non-Dirac heat pulse and also for variations in the input flux from the heating elements.

**2.2.2 Parameter Estimation**

Parameter estimation and uncertainty analysis was undertaken using the DREAM algorithm (Vrugt et al., 2009a; Vrugt et al.,
2009b). The fit of the data was performed by assessing the likelihood of each individual model run. The likelihood is defined
as:

$$L = -\left(\sum_{1}^{n_{obs}} \ln\left(p_{obs}\left(T_{mod} | T_{obs}, \sigma^2\right)\right) + \sum_{1}^{n_{pars}} \ln\left(p_{par}(X)\right)\right) \tag{17}$$





where $T_{\mathrm{mod}}$ and $T_{obs}$ represent the modelled and observed temperatures respectively, $\sigma^2$ represents the error of the temperature observation squared, $p_{obs}$ represents the probability of the modelled observation, assuming a normal distribution and a mean of $T_{obs}$ and a variance of $\sigma^2$, and $p_{par}$ represents the probability of the parameter $X$. We assume that the parameters are uniformly distributed, hence $p_{par}(X) = 1$ when the parameters are in bounds and zero elsewhere.

The DREAM method is an adaption of the standard Markov chain Monte Carlo method. The technique is initialised by specifying a number of chains. Each chain receives starting parameters by randomly sampling the parameter ranges (Table 1). After calculating an initial likelihood for the starting parameters, the algorithm selects proposed parameter values using the other chains, and the likelihood of these model parameters are also calculated. If the likelihood of these parameters is greater than the current parameters, the new parameters are accepted; however, if the likelihood of the new parameters is lower, the

transition probability is calculated using a ratio of the likelihoods, and the transition is determined by generating a random number. The method is explained in greater detail in Vrugt et al. (2009c). The general outcome is that the chains spend a greater amount of time in locations of more favourable parameters, and the distribution of these parameters represents the posterior distribution of the parameter probabilities, given the model, the data and the prior knowledge of the parameter distributions.

The optimisation was undertaken using five parameters: $\|\mathbf{q}\|$, $\theta$, $\phi$, $\kappa_0$ and $\rho c$. The default parameter likelihoods and initial distributions were taken from the ranges presented in Table 1. The error of the temperature observations ($\sigma$) was taken to be 0.06 °C; the precision of the temperature sensors. Whilst the model parameters are estimated using angle and the flow magnitude, the actual flow vector can be recovered as:

$$\mathbf{q} = \begin{bmatrix} \|\mathbf{q}\|\cos(\phi)\cos(\theta) \\ \|\mathbf{q}\|\cos(\phi)\sin(\theta) \\ \|\mathbf{q}\|\sin(\phi) \end{bmatrix} \tag{18}$$

**2.3 Laboratory Sand Tank**

A laboratory sand tank was used to provide a controlled environment on the hydraulic regime to test the performance of the Hot Rod. A total of 36 combinations of flow direction, magnitude and depth of heat pulse were used. The dimensions of the sand tank for each of the scenarios varied slightly according to the fixed boundary conditions (Figure 3). Four flow scenarios were tested: (1) horizontal flow from left to right (inflow and outflow occurred over the entire saturated cross sectional area of

the sediment volume on the left and right boundaries of the tank), (2) diagonal flow from the top left to bottom right (inflow was through a 20 mm horizontal inlet slit on the top left boundary and outflow through a 20 mm horizontal slit, 85 mm above the base of the right boundary), (3) upward flow (inflow occurred over the cross sectional area of the base of the tank and



outflow at the top of the sediment at overflow points above the sediment on the left and right boundaries, and (4) downward flow (inflow was distributed over the cross sectional area of the sediment surface and outflow via the cross sectional area of the tank base. A steady state flow regime for each scenario was maintained by constant heads at the inflow and outflow ports of the tank and the use of a peristaltic pump with a highly accurate ultrasonic flowmeter (Atrato ultrasonic flow meter; 0.05 %

linearity on flow less than 5 L min$^{-1}$) to ensure a constant discharge rate. The flowmeter data was also used to determine the Darcy flux for the different flow conditions. Fine perforated mesh was used to contain the sediment and to provide the necessary flow conditions along each of the tank boundaries. The hydraulic conductivity and thermal hydraulic conductivity of the graded, saturated sand were measured using a KSAT meter (UMS GmbH, Munich, Germany) and KD2Pro (Decagon, Washington, USA), respectively. Three different hydraulic gradients and discharge rates for the four flow scenarios were

conducted to capture low ($\sim 10^{-6}$ m s$^{-1}$), moderate ($\sim 10^{-5}$ m s$^{-1}$) and high flow conditions ($\sim 10^{-4}$ m s$^{-1}$). The Hot Rod was positioned in the middle of the sand tank with thermistor stick sensor number one orientated perpendicular (90 degrees) to the flow direction for all of the scenarios. To validate the spatial arrangement of the sensors, the horizontal flow scenario was repeated with thermistor stick sensor number one rotated to be 45 degrees to the direction of flow. In addition to these flow scenarios, a no-flow experiment was conducted to evaluate the analysis routine, where the boundary conditions meant that

there was stagnant water.

## 2.4 Experimental Field Site

The Sturt River, Adelaide, Australia is a perennial river system receiving the majority of its input from a wastewater treatment facility. The geomorphology of the river was characterised by a narrow channel, no more than 3 metres wide with 0.3–0.5 m deep sediment ranging from fine silt to coarse gravels overlying a tight, low-permeability clay. The selection of this field site

was part of another on-going investigation looking at attenuation of micro-pollutants in the hyporheic zone. The residence time in the hyporheic zone was critical in evaluating the stream attenuation modelling.

## 3 Results and Discussion

### 3.1 Laboratory Sand Tank

Overall, the modelled breakthrough curves closely fit the observed data from the 56 sensor array with the modelled curves

capturing the rising limb, peak and tail of the measured temperature data over the sample period (Figure 4). The variance of each parameter is included in the modelled temperature breakthrough curves; however, the uncertainty is so small that it cannot be seen without zooming in on the individual curves. Selected breakthrough curve plots are shown of the 4th vertical thermistor (158 mm depth) from each radial sensor location. Four flow scenarios are presented: (a) horizontal (Figure 4), (b) diagonal (Figure S1-a SI), (c) upwards (Figure S1-b SI) and (d) downwards (Figure S1-c SI). The tests presented were conducted at a

moderate flow rate. Thermistor 4 was 158 mm below the base plate and at a greater depth than the heat injection depth (R2; at 140 mm). Temperature increases associated with the heat pulse were observed at the inner sensor sticks (28 mm) more quickly



than at the outer (47 mm) sensor sticks in the horizontal flow scenario (cf. Figure 4-d & h). The inner sensors also displayed a steeper rising and falling limb compared to the outer sensors. The temperature response reflected the sensor position relative to the dominant flow direction. For example, sensor T3-4 (Figure 4-c) was directly in line and down gradient of the heat pulse, whilst sensor T7-4 (Figure 4-g) was in line but up-gradient of the heat pulse and therefore showed a smaller response. The

breakthrough curves from the diagonal flow scenario showed a similar response in those sensors up and down gradient of the heat pulse (Figure S1-a SI). In the upward and downward flow scenarios there was very little response in the outer thermistor sensors due to the dominant vertical component of flow (Figure S1-b to c SI). This low response of the outer sensors was even more pronounced under higher flow conditions in the upward and downward flow scenarios. The 3D time series videos showed clearly the migration of the heated plume vertically and highlighted the complexity of fitting multiple breakthrough curves to

the most likely solution (refer to SI).

Overall, the 3D flow fields calculated from the HPS Hot Rod in the laboratory sand tank for the four flux scenarios and heat injection depths (65, 140 and 215 mm) were consistent with the flow conditions established in the tank (Figure 5). Under left to right horizontal flow conditions (Figure 5- a/b), the modelled direction of flow from each of the heat injection depths is very similar with some slight offset to the observed flow in the y-direction, which was perpendicular to thermistor stick sensor 1

(90 degrees to the flow direction). There was also a slight deviation downwards in the z direction, particularly for the shallowest heat injection depth. To refute any bias in the optimisation routine and the array configuration, the Hot Rod was rotated by 45 degrees for the horizontal flow scenario and it showed a very similar output to when it was orientated 90 degrees to the flow direction.

Results from the other three scenarios showed that the modelled flux direction is close to parallel to the flow conditions

established in the tank and the magnitude of the flux was similar at each of the heat injection depths (Figure 5- c to h). Reviewing the time series data in the 3D plots (SI), the spreading of the heat pulse from the heat injection depth can be clearly detected indicating how the heat pulse moves along the established flow line. The thermistor highlighted in blue in the 3D plot was the sensor that showed the maximum temperature breakthrough curve and clearly shows a different orientation to the most likely flux direction (black arrow) as determined by the DREAM algorithm.

Some discrepancies in the direction of the modelled flow and differences in flux magnitude at each heat injection depth may be attributed to: (1) placement orientation and the angle of the sensor positions of the Hot Rod relative to the flow conditions established in the tank, (2) boundary conditions in the tank to establish flow, and (3) that the optimisation routine determines the best fit of all the observed data in a 3D volume around the heat injection depth.

The difference in flux magnitude at each heat injection depth may be attributed to the number of sensors that were used in the

optimisation routine. For example, at the heat injection depth, R2 (140 mm) there are 3 sensor arrays (an array being 8 sensors positioned horizontally around the central carbon fibre rod) vertically above R2 and 4 sensor arrays vertically below R2. In comparison, at heat injection depth, R1 (65 mm) there is only 1 sensor array vertically above R1 and 6 sensor arrays below R2, which may limit the optimisation routines for particular flow conditions i.e. strongly upwards flow.





In the case of no-flow conditions established in the sand tank (Figure S2. SI), the optimisation routine fitted the measured temperature breakthrough data, however, on closer inspection of the 3D time series plot it was evident based on the uniform heat plume around the heat injection point during the injection period that heat transport was by conduction only. Absence of clear advective movement of the heat pulse and a calculated flux less than about $\sim 10^{-6}$ m s$^{-1}$, indicated the lower limit of the active heat pulse sensor.

The flux magnitude ($\|\mathbf{q}\|$), of the different flow scenarios and different flow intensities calculated based on different heat injection depth of the HPS (grey bars) compared to the fluxes determined based on Darcys law (hydraulic gradient and hydraulic conductivity) and the measured discharge from the tank using an ultrasonic flow meter (Figure 6 and Table S2 SI). The measured saturated hydraulic conductivity according to the KSAT meter for the sand tank sand was $4.95 \times 10^{-4}$ m s$^{-1}$. The measured saturated thermal conductivity of the sand was 3 W m$^{-1}$ $^{\circ}$C$^{-1}$ (using the Decagon KD2Pro), whilst the average modelled $\kappa_0$ from all of the flow scenarios in the sand tank was 3.8 W m$^{-1}$ $^{\circ}$C$^{-1}$. The thermal conductivity is strongly influenced by the porosity, and therefore some differences can be expected due to changes to the particle density with the packing of the sediment, where by thermal conductivity increases with decreasing porosity (Smits et al., 2010).

Histograms of the flux magnitude ($\|\mathbf{q}\|$) and flux components in the $x$, $y$ and $z$ direction for the combination of most likely parameter values used in the DREAM algorithm were generated for each measurement. The results from heat injection depth, R2 for the horizontal flow scenario is shown in Figure S3 of the SI, which shows that the distribution is tightly constrained. This is also evident in cross correlation plots of the flux magnitude, thermal conductivity and the specific heat capacity (Figure S4 of the SI).

Constraining the range of the thermal conductivity values used in the optimisation routine to the known measured thermal conductivity of the sand from the KD2Pro instrument showed little impact on the calculated flux magnitude. However, comparing the modelled breakthrough curves from two optimisations when the range in thermal conductivity was limited to the measured known thermal conductivity (3 W m$^{-1}$ $^{\circ}$C$^{-1}$) in one model and in the other model it used a value of 3.52 W m$^{-1}$ $^{\circ}$C$^{-1}$ (for a best fit from a range of values from 2.5 to 4.5 W m$^{-1}$ $^{\circ}$C$^{-1}$) showed there was subtle differences between the modelled breakthrough curve peak and falling limbs (Figure 7).

## 3.2 Experimental Field Site

The measured 3D flow fields at the experimental site showed considerable variability in the direction of flow and flux magnitude over $\sim 0.20$ m depth of the streambed. The flux magnitude at the six stations along the river at the three heat injection depths (0.065, 0.140 and 0.215 m) ranged from $4.2 \times 10^{-6}$ to $4.26 \times 10^{-5}$ m s$^{-1}$ (mean: $1.6 \times 10^{-5}$ m s$^{-1}$). At each of the stations, the component of horizontal flow compared to vertical flow within the streambed was dominant. It was only at the shallowest heat injection depth, just below the streambed surface that there was a greater component of vertical flow. The results from two of the six stations is shown in Figure 8, where the Hot Rod was positioned in the streambed such that the x-axis of the figure was




aligned to the direction of stream flow. In such environments the flow direction is driven by the surface flow and the geomorphological features of the streambed. Small bed structures such as ripples will only impact the hyporheic flow field in the uppermost centimetres of the sediment which is a plausible explanation as to what was observed from the outputs of the Hot Rod.

Measured $K_0$ values for the sediment collected at the 6 stations from 0 to 0.2 m depth ranged from 0.9 to 2.84 W m$^{-1}$ $^\circ$C$^{-1}$, (mean: 2.07 W m$^{-1}$ $^\circ$C$^{-1}$) compared to the values that were determined by the optimisation routine, which ranged from 1.0 to 3.74 W m$^{-1}$ $^\circ$C$^{-1}$ (mean: 2.91 W m$^{-1}$ $^\circ$C$^{-1}$). Physical observation of the sediments collected at the experimental site stations showed that the streambed material was quite heterogeneous and also contained varying proportions of organic matter. Higher clay content and organic matter in the sediment would cause a higher volumetric heat capacity compared to sandy sediments.

The volumetric heat capacity also increases with increasing moisture content and particle density (Abu-Hamdeh, 2003;Barry-Macaulay et al., 2013;Jury and Horton, 2004). The assumption of uniform thermal properties of the 3D volume around the heat injection point is likely to contribute to the uncertainty in the flow direction. For example, Su et al (2006) used numerical simulations to show that differences in the thermal properties of the sediment around a flow sensor can lead to incorrect velocity estimates, and in particular differences in the horizontal and vertical flux components. Additional measurements of

the streambed sediments would provide a tighter constraint on the parameter set used in the optimisation; however, the parameter estimation and uncertainty analysis routine does successfully fit the measured data to provide a reliable estimate of the flux and its direction.

## 4 Conclusion

Despite the early pioneering work of Lewandowski et al. (2011) and Angermann et al. (2012b) for the concept of a 3D active

heat pulse sensor to determine flux and direction in the shallow streambed, their studies experienced a number of shortcomings that are related to the design of the instrument and the analysis of the data. This included: (1) a limited number of sensors and spatial positions around the heating element, (2) weakness with the sensor sticks wobbling and therefore poorly constrained sensor positions in relation to the heating element, (3) limited constraint on the input functions to the heat transport equation i.e. not knowing the current input, and (4) lack of a suitable optimisation routine to determine the most likely set of parameters

to constrain the data and an uncertainty analysis on the flux magnitude and its direction.

The rigidity and robustness of the Hot Rod and use of heating elements at three different vertical positions provided a method to examine how the flux and its direction varied vertically with depth beneath the streambed interface at individual locations in a range of different environmental settings and sediment types. The use of two horizontal spacings between the heating elements and thermistors as well as additional thermistors at multiple angles to the heating elements increased confidence in

the measurement of heat transport processes and tightened the optimisation routine of the temperature data. The addition of the measured input of energy at the heating element in the heat transport equation as a series of discrete heat pulses over the



injection period provided one less unknown variable to calibrate against. In many of the experiments conducted in the sand tank and at the experimental site, the optimisation routine using the DREAM algorithm showed that the most likely flux direction from the heat injection depth was not towards the sensor that showed the maximum temperature breakthrough because it uses all of the sensor temperature breakthrough curves in the analysis. The 3D time series plot was a valuable tool in assessing

this result and it also showed whether heat transport was dominated by diffusion/conduction with radial symmetry around the heating element or that there was convective heat flow. This interrogation process was found to be critical in the data assessment to ensure that the model did not over-fit the measured data with unrealistic physical values for the sediment and heat transport conditions.

The laboratory and experimental field site applications using the DREAM algorithm for parameter estimation and uncertainty

analysis demonstrated the performance of the active heat pulse sensing instrument (the Hot Rod) to measure the multi-directional 3D-flow fields and fluxes in the near surface streambed. Active heat pulse sensing provides a number of advantages over other approaches that have investigated hyporheic exchange, including the low cost of data collection and the rapid assessment of small physical processes that can be undertaken at a reach scale. Marzadri et al. (2013) showed that the hyporheic residence time, which is influenced by the streambed physical morphology and in-stream flow discharge, ultimately determines

the spatially complex patterns of the time varying thermal regime within the hyporheic zone. The short duration active heat pulse sensing helped overcome some of the challenges in measuring the water temperatures because of the stronger signal from the heat pulse.

Most other studies that use heat as a tracer assume 1D flow only and the lateral or horizontal component is not considered. Studies that have identified the geometry of the subsurface flow field using a polynomial model fitted to the amplitude ratio

of the vertical temperature profiles were only able to determine the deviation from one-dimensional vertical flow (Munz et al, 2016). Errors in the vertical component of flow have been shown to progressively increase with the magnitude of the horizontal flow component (Lautz, 2010). The 3D analysis routine and sensor arrangement applied in this study was able to capture all three components of the flow field around the point of observation. The importance of capturing the multi-directional flow field was clearly demonstrated in the sand tank under an extensive range of flow conditions that would be anticipated in a

dynamic stream environment. Measurements of the hydraulic gradient and characterising the physical properties of the streambed sediment are also important in understanding the dynamic exchange processes within the hyporheic zone and the very transient nature of such environments.

### Acknowledgments

We are grateful to Flinders University South Australia for a small grant to develop the HPS Hot Rod and all Faculty of Science

and Engineering technical workshop staff for their assistance in hardware development and construction. Additional funding through the Australia-Germany Joint Research Cooperation Scheme of Universities Australia and the German Academic Exchange Service (DAAD, grant no. 57216806) provided support for fieldwork collaboration between the research institutes.



## Supporting Information

Contains details of Hot Rod sensor array, results from the Hot Rod tests conducted in the sand tank, including selected breakthrough curve plots and a short video compilation showing the 3D time series data of the sand tank flow scenarios.

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





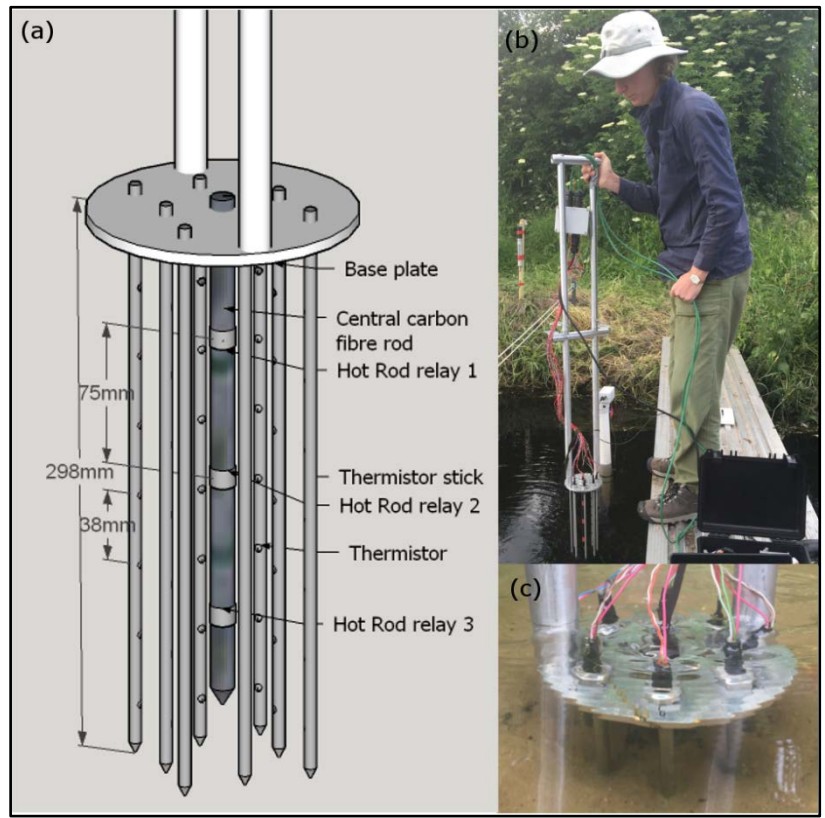

**Figure 1: (a) Detailed design of the active HPS Hot Rod with 3 heating elements on the central carbon fibre rod surrounded by 56 temperature sensors at two distances from the central heat source (28 and 47 mm). (b) and (c) Installation of the Hot Rod in a small stream characterised by shallow bedforms.**

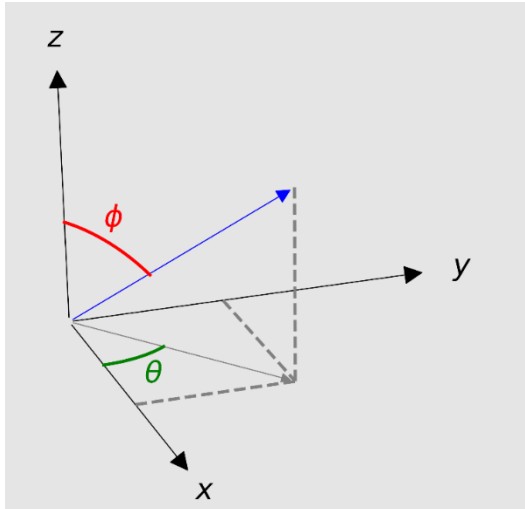

**Figure 2: Schematic showing the rotation of the coordinate system to determine angles $\theta$ and $\phi$.**





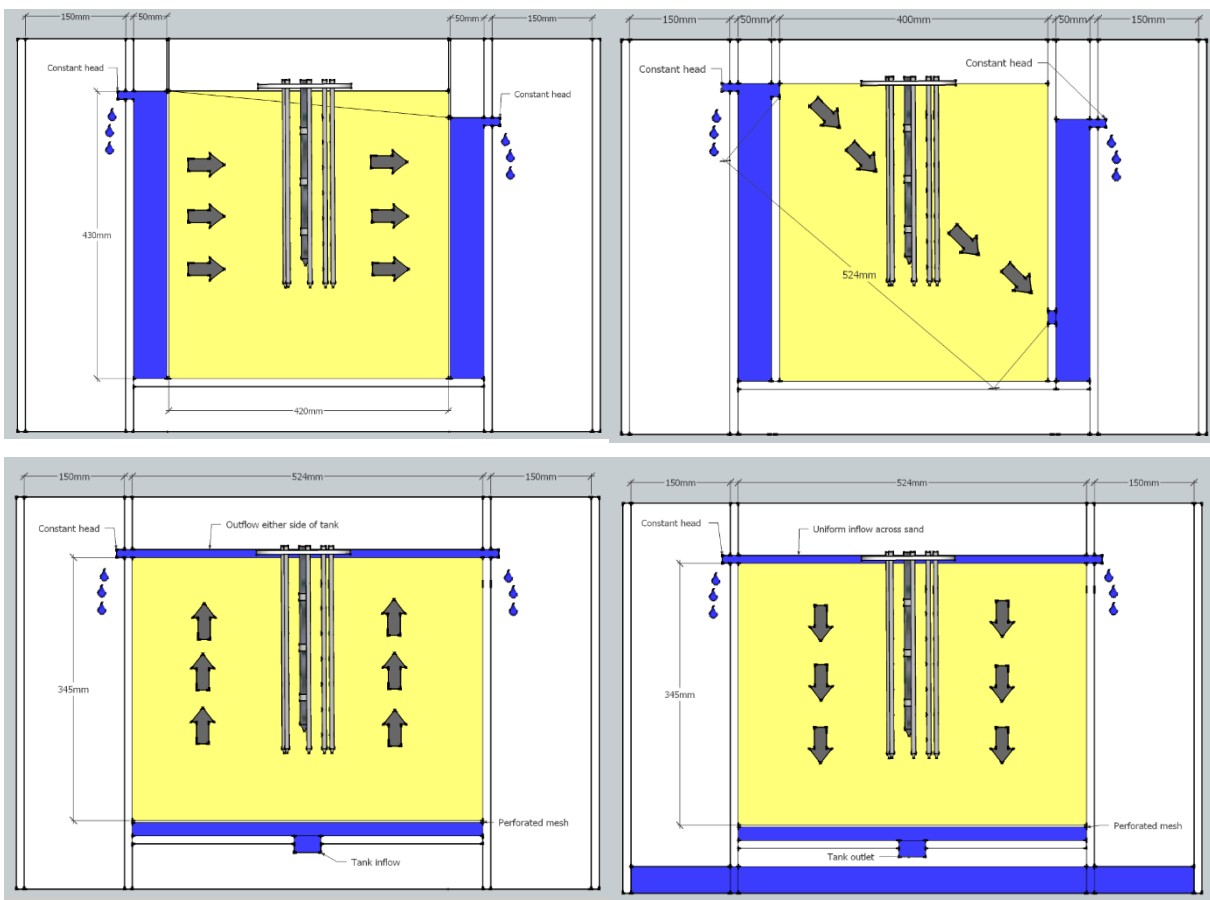

**Figure 3: Laboratory sand tank dimensions for each of the flow scenarios (a) horizontal flow, (b) diagonal flow, (c) upward flow,**
5  **and (d) downward flow.**





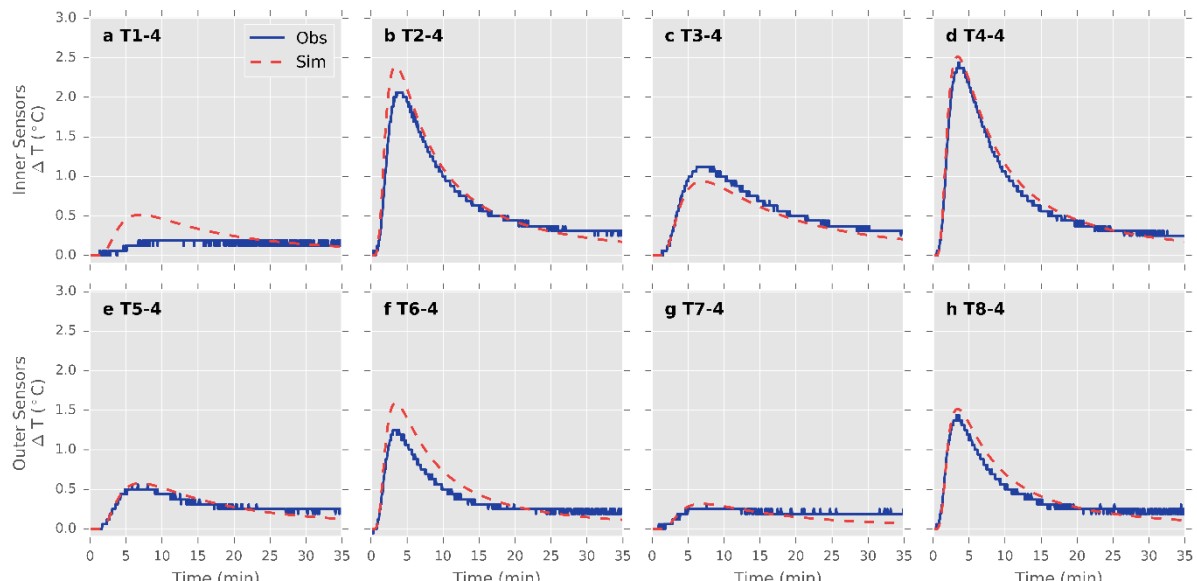

**Figure 4: Breakthrough curves shown of the 4th vertical thermistor (158 mm depth) from each radial sensor location for the horizontal flow scenario from heat injection depth at relay 2. Solid lines are observed and dashed lines are modelled.**





Figure 5: Calculated fluxes and directions for the four flow scenarios at each of the heat injection depths: (a-b) horizontal, (c-d) diagonal, (e-f) upward and (g-h) downward flow scenarios.





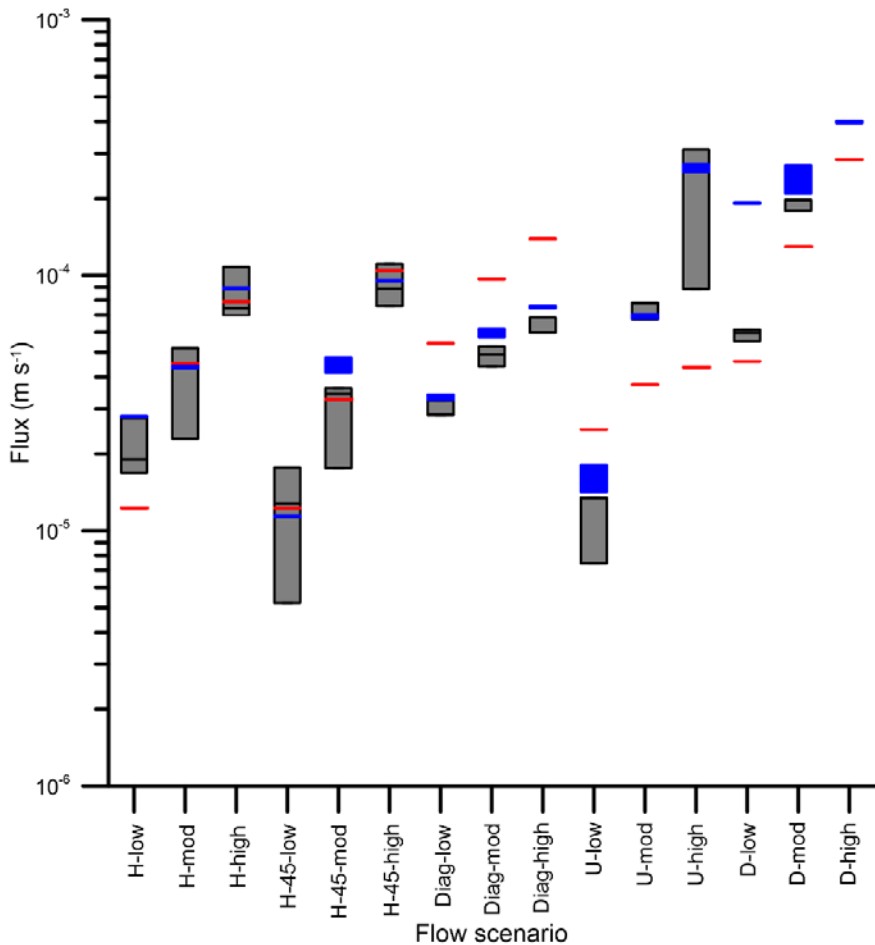

**Figure 6: Fluxes of the different flow scenarios and different flow intensities calculated based on different heat injection depth of the HPS (grey bars) compared to fluxes determined based on Darcys law (hydraulic gradient and hydraulic conductivity- red dashes) and the measured discharge from the tank using an ultrasonic flow meter (blue dashes).**




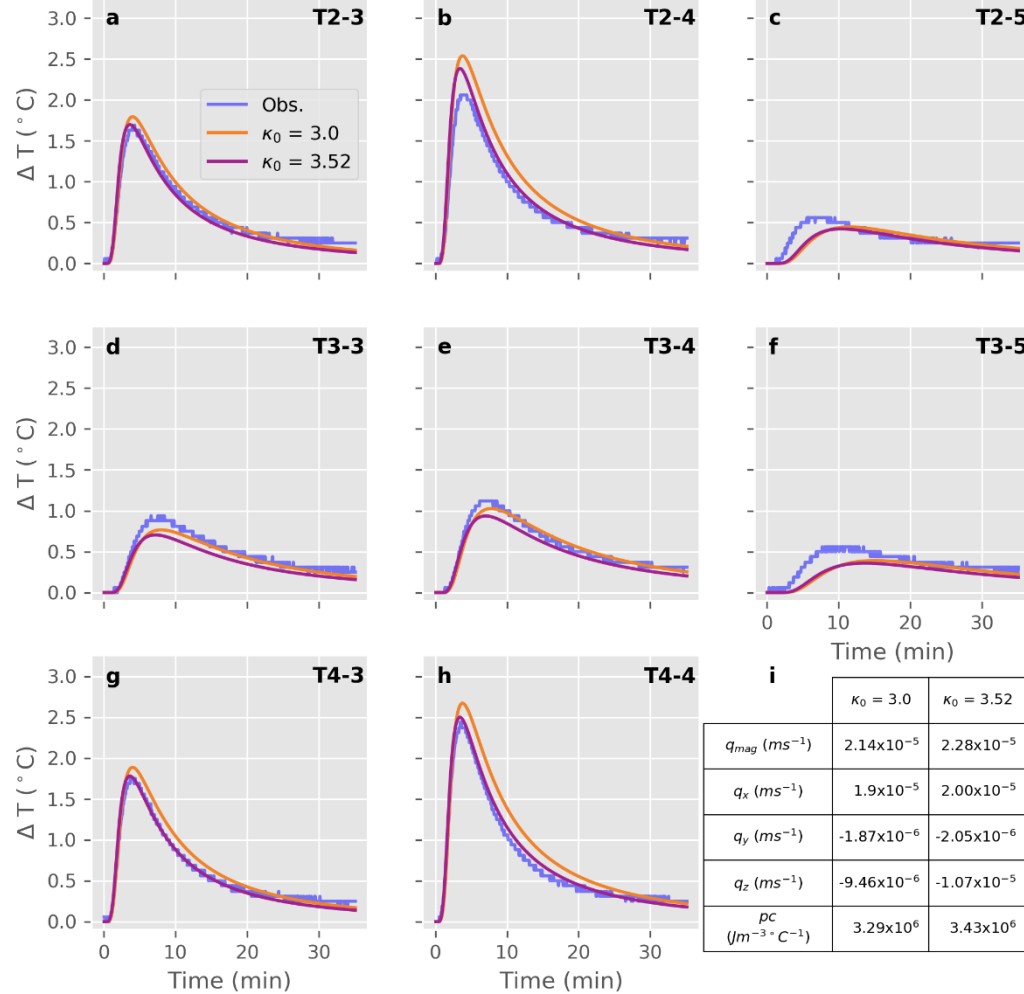

**Figure 7: Comparison of the observed (blue line) and modelled breakthrough curves for selected temperature sensors (Tx-x) when the thermal conductivity, $\kappa_0$ of the sediment has a known value of 3 W m$^{-1}$ $^{o}$C$^{-1}$ (orange graph) and when the model is provided with a range of plausible values from 2.5 to 4.5 W m$^{-1}$ $^{o}$C$^{-1}$ resulting in 3.52 W m$^{-1}$$^{o}$C$^{-1}$ (red graph).**





**Figure 8: Calculated fluxes and flux directions at the three heat injection depths from two of the stations at the experimental field site. (Left) Station 1 and, (Right) Station 2. The x-axis in the figures is positive in the direction of streamflow.**





**Table 1. Initial parameter values for $\|\mathbf{q}\|$, $\theta$, $\phi$, $\mathcal{K}_0$ and $\rho c$ used in the analysis routine. A uniform distribution of the 5 parameters was used.**

| Parameter | Mean | Width |
|---|---|---|
| $\|\mathbf{q}\|$-magnitude (m s$^{-1}$) | $10^{-5}$ | $10^{3}$ |
| $\mathcal{K}_0$ (W m$^{-1}$ °C$^{-1}$) | 3 | 1.25 |
| $\rho c$ (J m$^{-3}$ °C$^{-1}$) | 2750000 | 1500000 |
| $\theta$ | $\pi$ | $2\pi$ |
| $\phi$ | $\pi$ | $2\pi$ |