# Peer review of "Active heat pulse sensing of 3D-flow fields in streambeds"

_Hydrology and Earth System Sciences, 2017_

## Referee Comment (RC1) · J. Constantz (Referee) · 15 Nov 2017

This manuscript represents a very nice scientific piece of work, describing an innovative heat-tracing tool to track 3-D shallow groundwater flow that's especially useful in streams, but also useful such hydrologic settings as hillslopes with measurable shallow interflow (which the authors might consider mentioning in there final draft). Itemize comments are below: 1. This review assumes Eq. [6] thru [18] are correct, since this is not this reviewers field of professional interest. 2. A summary of the extensive use of down-hole thermal pulse techniques in groundwater flow studies should be included, focusing on these equipment, methods and results. 3. Discussion in this manuscript jumps from 1-D o 3-D streambed flow patterns, without any mention of several good 2-D studies using heat as a tracer, such as, Constantz et al. (WRR, 2013) and Constantz et al. (WRR, 2016). 4. Finally, all the figures are very nice; however, in Figure 3, visually its unclear why there is upward flux in the specific drawing to represent this direction of flow, which requires a significant hydraulic head to create this flow pattern. Is it possible to show this in the figure itself, in addition to explaining in the text?

---

## Referee Comment (RC2) · B. des Tombe (Referee) · 13 Dec 2017

**General comments** The authors introduce a new device to measure flow in the hyporheic zone three depths with an active heat tracer experiment. The authors share sufficient knowledge about their device and the design choices. The device was first tested in a lab-environment with good results, after which it was used in a field case. I hereby recommend this article to be published, however I do have a comment on the formula used and a few suggestions.

**Specific comments** Equation 4 is the impulse response. To end up with a pulse response, Eq4 needs to be integrated over time to end up with a step response. Subtract a shifted step response from a not-shifted step response to end up with a pulse re-

sponse. For example, see Bakker et al. 2015. The 3D step response, is quite difficult to derive, but is done by [Hunt 1983, Mathematical analysis of groundwater resources], see Equation 6.53, page 135. And is used in [Rau et al. 2012, Experimental investigation of the thermal dispersivity term and its significance in the heat transport equation for flow in sediments], Equation 13. Stretching an impulse response to a pulse response, can be done if the width of the pulse is much smaller than the response duration, which is not the case here I would say. The pulse width here is 1 minute and most of the dynamics/response appear within the first 10 minutes. Furthermore, the units on the right hand side of Equation 4 are in $°Cs^{-1}$.

The effects of hydrodynamic dispersion are neglected/not mentioned throughout the entire analysis. There is plenty of research done that suggests that with these velocities, hydrodynamic dispersion does play part. Make sure it can be neglected before you neglect is. For example [Rau et al. 2012, Experimental investigation of the thermal dispersivity term and its significance in the heat transport equation for flow in sediments].

I would like to see a back-of-the-envelope calculation that the tank in the lab setup is sufficiently large and the boundaries do not affect the uniformity of the flow. Especially in the diagonal flow situation. This might end up being one or two lines of text in the article, but important.

**Technical corrections**
P2L4: What makes it challenging to determine the non-vertical component?
P2L16: Kind of vague, 'more robust' and 'advanced analysis'. (there is a 'the' too much)
P2L28: R1, R2, and R3 are not listed as such in Figure 1.
P3L27: Pore flow velocity is never being calculated; there is no porosity in the entire formulation.
P4 Equation 3: Not needed perse

P5 Equation 6: For the original formulation with the impulse response, the $q_y$ and $q_z$ could have been put directly in the exponent of the equation.

P6L5-7: Be more specific, rephrase. Also, see previous comment.

P6L11: Be specific which boundary condition you may variate.

P6 Equation 16: see first of the specific comments.

P7L1: some subscripts are italic and others not.

P9L27: boundary conditions in the tank to establish flow. The great advantage of having a lab setup is that all boundary conditions are under control. This should be an important aspect of that section.

P10L5:active heat pulse sensor. try to be consistent with the naming of the device.

---

## Referee Comment (RC3) · B. des Tombe (Referee) · 13 Dec 2017

Figure 6: To me it is unclear how the upper and lower bounds of the grey bars are calculated. As a suggestion, maybe link the x-axis tick labels to the ones in Fig.3.

Figure 8: I would suggest to just list the magnitudes of the arrows in a legend. The m for meters and s for seconds should not be italic.

[Figure]

---

## Author Comment (AC1) · 19 Jan 2018

\*\*Note: Reviewer questions are shown in black text, response to reviewers comments in blue text with changes in manuscript shown in red text in the attached pdf file.

Reviewer 1- J. Constantz This manuscript represents a very nice scientific piece of work, describing an innovative heat-tracing tool to track 3-D shallow groundwater flow that's especially useful in streams, but also useful such hydrologic settings as hillslopes with measurable shallow interflow (which the authors might consider mentioning in their final draft).

Good suggestion. Have added a sentence in the conclusion section: 'The concept and design of the active heat pulse sensing instrument could also be adapted to other

hydrological research areas, including the measurement of shallow interflow along hill-slopes and discharge from groundwater seeps and springs.'

Reviewer specific comments

1. This review assumes Eq. [6] thru [18] are correct, since this is not this reviewer's field of professional interest. The equations have been revised based on suggestions from reviewer 2.

2. A summary of the extensive use of down-hole thermal pulse techniques in ground-water flow studies should be included, focusing on these equipment, methods and results.

The authors did not focus extensively on the historical development of thermal pulse techniques as this was covered in greater depth by Lewandowski et al. 2011. The technique has been widely used in downhole applications, particularly with the use of distributed temperature sensing and a number of references have been referred to in the manuscript. To keep the focus of the paper on examining the 3D flow processes in the streambed and the design of the instrument, the authors do not see that it is necessary to provide any further additional summary of down-hole thermal pulse techniques within the literature review.

3. Discussion in this manuscript jumps from 1-D to 3-D streambed flow patterns, without any mention of several good 2-D studies using heat as a tracer, such as, Constantz et al. (WRR, 2013) and Constantz et al. (WRR, 2016).

Yes. Good point. We wanted to highlight the contrast between the interpretation of 1D and 3D data. 3D data is also often distilled down to a 2D array when there is a dominant horizontal direction of flow. Reference to the use of vertical temperature profiles along transects to examine 2D flow fields in the streambed has been included in the revised manuscript.

Added to the introduction section (P1L30): 'Series of vertical temperature profile sticks

installed along transects have also be used in other studies to examine 2D flow fields in the streambed (Constantz et al., 2013;Constantz et al., 2016;Shanafield et al., 2010).'

4. Finally, all the figures are very nice; however, in Figure 3, visually it's unclear why there is upward flux in the specific drawing to represent this direction of flow, which requires a significant hydraulic head to create this flow pattern. Is it possible to show this in the figure itself, in addition to explaining in the text?

In Figure 3 there are subtle differences in the arrangement of the sand tank arrangement for the different flow scenarios. The upwards flux scenario (Figure 3c) is different to the downwards flux scenario (Figure 3d) by where the water enters and exits the tank as described by the words in the figure with "inflow" and "outflow" at the top and lower boundaries. Figure 3 (c) is upwards as was enters via the inlet at the tank base and exits at the top boundary via the constant head overflow points (the upwards head gradient was maintained via a hose attached to the inlet at the base of the sand tank); on the other hand, Figure 3 (d) is downwards with flow entering across the top boundary and exiting via the outlet at the tank base. Arrows and additional text have been added to the figure to provide some further clarity.

Please also note the supplement to this comment:
https://www.hydrol-earth-syst-sci-discuss.net/hess-2017-582/hess-2017-582-AC1-supplement.pdf

―――――――――――――――

**Supplement:**

**Active heat pulse sensing of 3D-flow fields in streambeds**

Eddie W. Banks* [1], Margaret A. Shanafield [1], Saskia Noorduijn [1], James McCallum [1], Jörg Lewandowski [2,3], Okke Batelaan [1]

[1] National Centre for Groundwater Research and Training and the College of Science and Engineering, Flinders University, Adelaide, South Australia, Australia.

[2] IGB, Leibniz-Institute of Freshwater Ecology and Inland Fisheries, Department Ecohydrology, Berlin, Germany.

[3] Humboldt University Berlin, Geography Department, Berlin, Germany.

*Correspondence to*: Eddie W Banks (eddie.banks@flinders.edu.au)

**\*\*Note: Reviewer questions are shown in black text, response to reviewers comments in blue text with changes in manuscript shown in red text.**

**Reviewer 1- J. Constantz**

This manuscript represents a very nice scientific piece of work, describing an innovative heat-tracing tool to track 3-D shallow groundwater flow that's especially useful in streams, but also useful such hydrologic settings as hillslopes with measurable shallow interflow (which the authors might consider mentioning in their final draft).

Good suggestion. Have added a sentence in the conclusion section: 'The concept and design of the active heat pulse sensing instrument could also be adapted to other hydrological research areas, including the measurement of shallow interflow along hillslopes and discharge from groundwater seeps and springs.'

**Reviewer specific comments**

1. This review assumes Eq. [6] thru [18] are correct, since this is not this reviewer's field of professional interest.

The equations have been revised based on suggestions from reviewer 2.

2. A summary of the extensive use of down-hole thermal pulse techniques in groundwater flow studies should be included, focusing on these equipment, methods and results.

The authors did not focus extensively on the historical development of thermal pulse techniques as this was covered in greater depth by Lewandowski et al. 2011. The technique has been widely used in downhole applications, particularly with the use of distributed temperature sensing and a number of references have been referred to in the manuscript. To keep the focus of the paper on examining the 3D flow processes in the streambed and the design of the instrument, the authors do not see that it is necessary to provide any further additional summary of down-hole thermal pulse techniques within the literature review.

3. Discussion in this manuscript jumps from 1-D to 3-D streambed flow patterns, without any mention of several good 2-D studies using heat as a tracer, such as, Constantz et al. (WRR, 2013) and Constantz et al. (WRR, 2016).

Yes. Good point. We wanted to highlight the contrast between the interpretation of 1D and 3D data. 3D data is also often distilled down to a 2D array when there is a dominant horizontal direction of flow. Reference to the use of vertical temperature profiles along transects to examine 2D flow fields in the streambed has been included in the revised manuscript.

5        Added to the introduction section (P1L30): 'Series of vertical temperature profile sticks installed along transects have also be used in other studies to examine 2D flow fields in the streambed (Constantz et al., 2013;Constantz et al., 2016;Shanafield et al., 2010).'

4. Finally, all the figures are very nice; however, in Figure 3, visually it's unclear why there is upward flux in the specific drawing to represent this direction of flow, which requires a significant hydraulic head to create this flow pattern. Is it possible
10    to show this in the figure itself, in addition to explaining in the text?

In Figure 3 there are subtle differences in the arrangement of the sand tank arrangement for the different flow scenarios. The upwards flux scenario (Figure 3c) is different to the downwards flux scenario (Figure 3d) by where the water enters and exits the tank as described by the words in the figure with "inflow" and "outflow" at the top and lower boundaries. Figure 3 (c) is upwards as was enters via the inlet at the tank base and exits at the top boundary via
15    the constant head overflow points (the upwards head gradient was maintained via a hose attached to the inlet at the base of the sand tank); on the other hand, Figure 3 (d) is downwards with flow entering across the top boundary and exiting via the outlet at the tank base. Arrows and additional text have been added to the figure to provide some further clarity.

**Reviewer 2- B. des Tombe**

**General comments**

The authors introduce a new device to measure flow in the hyporheic zone at three depths with an active heat tracer experiment. The authors share sufficient knowledge about their device and the design choices. The device was first tested in a lab-environment with good results, after which it was used in a field case. I hereby recommend this article to be published, however I do have a comment on the formula used and a few suggestions.

**Specific comments**

1. Equation 4 is the impulse response. To end up with a pulse response, Eq4 needs to be integrated over time to end up with a step response. Subtract a shifted step response from a not-shifted step response to end up with a pulse response. For example, see Bakker et al. 2015. The 3D step response, is quite difficult to derive, but is done by [Hunt 1983, Mathematical analysis of groundwater resources], see Equation 6.53, page 135. And is used in [Rau et al. 2012, Experimental investigation of the thermal dispersivity term and its significance in the heat transport equation for flow in sediments], Equation 13. Stretching an impulse response to a pulse response, can be done if the width of the pulse is much smaller than the response duration, which is not the case here I would say. The pulse width here is 1 minute and most of the dynamics/response appear within the first 10 minutes. Furthermore, the units on the right hand side of Equation 4 are in _Cs-1.

*The manuscript section describing the heat transport simulation (2.2.1) has been revised in the manuscript to account for the thermal dispersivity term and also to clarify the impulse response function and how the known input current from the heating elements was used in the pulse response calculation. The units on the right hand side of Equation 4 have also be rectified. The amendments made to the mathematical equations in the revised manuscript are described below:*

The magnitude and direction of the water velocity at the observation point based on the measured temperature breakthrough curves at the 56 sensors was determined using a modified version of the heat transport equation:

$$\frac{\partial T}{\partial t} = \nabla\left(D^t \nabla T\right) - \nabla\left(\frac{\rho_w c_w}{\rho c} q T\right) \tag{1}$$

where $T$ is temperature (°C), $D^t$ is the thermal dispersion coefficient given as (de Marsily, 1986):

$$D_n^t = \frac{\kappa_0}{\rho c} + \beta_n \cdot \left|\frac{\rho_w c_w}{\rho c} q\right| \tag{2}$$

$\kappa_0$ is the bulk thermal conductivity of the water saturated sediments (W m$^{-1}$ °C$^{-1}$), $\rho c$ is the volumetric heat capacity of the water saturated sediments (J m$^{-3}$ °C$^{-1}$), $\beta_n$ is the thermal dispersivity where the subscript $n$ is $T$ for the transverse direction

and *L* for the longitudinal direction, $\rho_w c_w$ is the volumetric heat capacity of water (4.1 J m$^{-3}$ $^{\circ}$C$^{-1}$) and $q$ is the Darcy velocity

(or Darcy flux) of water (m s$^{-1}$). Where $D_L^t \approx D_T^t$, equation 2 can be simplified to $D_n^t = \dfrac{\kappa_0}{\rho c}$.

An analogy can be made to the solute transport equation where the mean water velocity is replaced with $\dfrac{\rho_w c_w}{\rho c} q$ and the

dispersion tensor can be replaced with $D^t$. Assuming that these components are constant in space and time, Equation 1 can

5  be reduced to:

$$\frac{\partial T}{\partial t} = D^t \nabla^2 T - \frac{\rho_w c_w}{\rho c} q \nabla T \tag{3}$$

For an instantaneous injection of a thermal mass into an infinite three-dimensional sediment where there is only an $x$

component of velocities can be given as (Domenico and Schwartz, 1998):

10  $$T(x,y,z,t) = \frac{M_0}{8\rho c D_L^{t\,\frac{1}{2}} D_T^t (\pi t)^{\frac{3}{2}}} \exp\left( -\frac{\left(x - \dfrac{\rho_w c_w}{\rho c} qt\right)^2}{4 D_L^t t} - \frac{\left(y^2 + z^2\right)}{4 D_T^t t} \right) \tag{4}$$

where $M_0$ is the thermal mass (J).

Equations 5 through to 14 stay the same and is followed by:

15  We then substitute these new dimensions into Equation 4 to get:

$$T(x'',y'',z'',t) = \frac{M_0}{8\rho c D_L^{t\,\frac{1}{2}} D_T^t (\pi t)^{\frac{3}{2}}} \exp\left( -\frac{\left(x'' - \dfrac{\rho_w c_w}{\rho c} \|q\| t\right)^2}{4 D_L^t t} - \frac{\left(y''^2 + z''^2\right)}{4 D_T^t t} \right) \tag{15}$$

Equation 15, represents an impulse response function. The tests implemented used a finite pulse that did not meet this

condition. As we assume that the properties are not temperature dependent, we can treat the contribution of multiple impulse

responses as additive. Hence, Equation 15 was implemented for a series of discrete, lagged pulses to represent the actual

20  addition of thermal mass to the system. The use of discrete pulses is implemented as:

$$T_{tot}(x'', y'', z'', t) = \sum_{\tau=t_{on}}^{t_{off}} T(x'', y'', z'', t-\tau) \qquad (16)$$

where $T_{tot}$ is the total temperature response, $t_{on}$ is the time at which the heating element was turned on and $t_{off}$ is the time

when the heating element was turned off. The operator $\tau$ represents the time lags, and for each discrete pulse $M_o = F \cdot d\tau$

. The summed $T$ term on the RHS is evaluated using Equation 15 for $t \geq \tau$, and is zero otherwise. This method allows for the

representation of a non-Dirac heat pulse and also for variations in the input flux from the heating elements. The interval $d\tau$

was 3 seconds, hence the Dirac representation was valid at the small scale.

2. The effects of hydrodynamic dispersion are neglected/not mentioned throughout the entire analysis. There is plenty of research done that suggests that with these velocities, hydrodynamic dispersion does play part. Make sure it can be neglected before you neglect it. For example [Rau et al. 2012, Experimental investigation of the thermal dispersivity term and its significance in the heat transport equation for flow in sediments].

The thermal dispersivity coefficient, $D^t$, representing both the longitudinal and transverse thermal dispersivity was included in the original heat transport simulation of the study, however, it was found not to make a significant difference to the calculated fluxes and therefore was not included in the final results (less than 2% difference in calculated flux with dispersion than without- see table below).

| Flow scenario | Horizontal_R1 | Horizontal_R2 | Horizontal_R3 | Diagonal_R1 | Diagonal_R2 | Diagonal_R3 |
|---|---|---|---|---|---|---|
| Mean Q (m/s) With dispersion | 5.21E-05 | 2.30E-05 | 5.22E-05 | 4.73E-05 | 4.39E-05 | 5.32E-05 |
| Mean Q (m/s) No dispersion | 5.20E-05 | 2.30E-05 | 5.24E-05 | 4.86E-05 | 4.40E-05 | 5.27E-05 |
| Difference in Q | 8.78E-08 | 1.06E-08 | 2.02E-07 | 1.30E-06 | 6.79E-08 | 5.60E-07 |
| % difference | 0.17 | 0.05 | 0.39 | 2.68 | 0.15 | 1.06 |

The description and equations in the heat transport simulation section of the manuscript have been revised to include the dispersion term to be consistent with other studies (i.e. Rau, 2012).

Revised text includes 'Our study found that the inclusion of longitudinal and transverse thermal dispersion had less than two percent difference on the mean calculated fluxes. The study by Rau et al. (2012) determined Darcy velocities derived from heat experimentation that included the thermal dispersivity term differed by up to 20 % when compared to solute experimentation. However, other studies in the literature have shown that there is considerable uncertainty on the magnitude of the thermal dispersivity (Anderson, 2005). Thermal dispersivity has also been found not to be scale dependant like solute dispersivity because heat transport happens through the pore water and through the sediment matrix (Vandenbohede et al., 2009). Therefore, given the scale that we are working at (few centimetres) and also the low velocities, the effect on the calculated flux is likely to be negligible.'

3. I would like to see a back-of-the-envelope calculation that the tank in the lab setup is sufficiently large and the boundaries do not affect the uniformity of the flow. Especially in the diagonal flow situation. This might end up being one or two lines of text in the article, but important.

The sand tank dimensions, physical characteristics of the sand and established flow conditions were implemented in a Hydrus model. The flow vectors from the Hydrus model showed that there was no significant impacts of the boundary conditions of the sand tank setup on where the hot rod was positioned in the tank. The figure R1 below shows the horizontal and diagonal flow scenario for the sand tank. Modelled results from each of the relay depths (R1, R2, R3) of the Hot Rod did vary slightly as a result of their depth position; more so for the diagonal flow field.

Figure 6 in the manuscript also shows a comparison of the fluxes calculated from the Hot Rod from the 3 relay depths represented by the grey bars and those based on Darcy's law (red dashes). The reported values are also shown in Table 2 of the Supporting Information.

[Figure]

Figure R1. Hydrus model of horizontal (top) and diagonal flow scenarios (bottom). Vertical red line represents the central vertical axis of the Hot Rod which was positioned in the middle of the sand tank.

**Technical corrections**

P2L4: What makes it challenging to determine the non-vertical component?

> Most applications in the sw-gw field that have used temperature have assumed 1D flow because thermistor sticks are deployed with a series of sensors in a vertical array and there are a number of routine analytical solutions and software packages to process the data. A 2D or 3D array is much more challenging with the physical installation of sensors to measure the flow field as well as the mathematical framework to process the data. This study demonstrates an application of a 3D sensor array and also an optimization routine to process the data to evaluate the 3D flow field in the shallow subsurface.

> Have revised the sentence to: "There are very few investigations which have tried to capture both the vertical and horizontal component of flow, as the determination of the non-vertical component is challenging with the physical installation of sensors to measure the flow field as well as the mathematical framework to process the data (Munz et al., 2016;Briggs et al., 2012;Shanafield et al., 2016)."

P2L16: Kind of vague, 'more robust' and 'advanced analysis'. (there is a 'the' too much)

> Added: more 'physically' robust. The reference to 'advanced analysis is based on the fact that this was not undertaken or little detail was provided on the analysis routine that was mentioned in Angermann et al. 2012. Removed 'the' from before the word 'temperature'.

P2L28: R1, R2, and R3 are not listed as such in Figure 1.

> R1, R2 and R3 labels have been added to Figure 1 in parentheses.

P3L27: Pore flow velocity is never being calculated; there is no porosity in the entire formulation.

> Correct. Should just be water velocity. Have amended the two locations in the manuscript where this was overlooked.

P4 Equation 3: Not needed perse

> Removed.

P5 Equation 6: For the original formulation with the impulse response, the qy and qz could have been put directly in the exponent of the equation.

> The reviewer is correct and these terms were initially removed to include dispersion, however, we went on to represent this equation only in terms of thermal conductivity. We have added flow components of the dispersion tensor back to this equation in addressing the reviewers other comments which now requires that the flow is expressed only in terms of an x-component.

Text has been modified prior to equation 6 to include: "The requirement of Equation 4 is that the flow component is only in the x-direction; removing the non-diagonal components of the dispersion tensor. The aim of this paper is to define a flow direction and magnitude using a fixed sensor array, allowing the use of multiple sensors to constrain the thermal transport and flow properties. It is not always the case that the fluxes are oriented with the sensor array and therefore the location of the observations can be converted through rotation of the coordinate system, aligning the measurements relative to the flow direction."

P6L5-7: Be more specific, rephrase. Also, see previous comment.

We have now explained the orientation of the sensor array in relation to the flow direction, and justified its necessity when using full dispersion terms. See revised changes to the manuscript in response to previous comments.

P6L11: Be specific which boundary condition you may variate.

This is addressed in the previous comments and the revised changes to manuscript.

P6 Equation 16: see first of the specific comments.

This comment has been addressed above in specific comment 1 describing how the impulse response function was treated.

P7L1: some subscripts are italic and others not.

Have corrected subscripts that were not italic.

P9L27: boundary conditions in the tank to establish flow. The great advantage of having a lab setup is that all boundary conditions are under control. This should be an important aspect of that section.

Yes we agree that in the laboratory setup all boundary conditions are under control to establish the desired flow conditions in the tank and was necessary so that we could compare flux calculation methods.

Have added to section 2.3: 'A laboratory sand tank was used to provide a controlled environment on the hydraulic regime to test the performance of the Hot Rod and so that the different flux calculation methods could be compared.'

The packing of the uniform graded sand also has some impact on flow conditions and we did our best to create an isotropic "streambed" with the active HPS instrument installed within the central position of the tank.

P10L5: active heat pulse sensor. try to be consistent with the naming of the device.

Replaced "active heat pulse sensor with" "active heat pulse sensing instrument". Also replaced at P2L15.

References

Anderson, M. P.: Heat as a ground water tracer, Ground Water, 43, 951-968, 2005.

Briggs, M. A., Lautz, L. K., McKenzie, J. M., Gordon, R. P., and Hare, D. K.: Using high-resolution distributed temperature sensing to quantify spatial and temporal variability in vertical hyporheic flux, Water Resources Research, 48, n/a-n/a, 10.1029/2011WR011227, 2012.

Constantz, J., Eddy-Miller, C. A., Wheeler, J. D., and Essaid, H. I.: Streambed exchanges along tributary streams in humid watersheds, Water Resources Research, 49, 2197-2204, 10.1002/wrcr.20194, 2013.

Constantz, J., Naranjo, R., Niswonger, R., Allander, K., Neilson, B., Rosenberry, D., Smith, D., Rosecrans, C., and Stonestrom, D.: Groundwater exchanges near a channelized versus unmodified stream mouth discharging to a subalpine lake, Water Resources Research, 52, 2157-2177, 10.1002/2015WR017013, 2016.

de Marsily, G.: Quantitative hydrogeology : groundwater hydrology for engineers, Academic Press, 1986.

Domenico, P. A., and Schwartz, F. W.: Physical and chemical hydrogeology, v. 1, Wiley, 1998.

Munz, M., Oswald, S. E., and Schmidt, C.: Analysis of riverbed temperatures to determine the geometry of subsurface water flow around in-stream geomorphological structures, Journal of Hydrology, 539, 74-87, http://dx.doi.org/10.1016/j.jhydrol.2016.05.012, 2016.

Rau, G. C., Andersen, M. S., and Acworth, R. I.: Experimental investigation of the thermal dispersivity term and its significance in the heat transport equation for flow in sediments, Water Resources Research, 48, n/a-n/a, 10.1029/2011WR011038, 2012.

Shanafield, M., Pohll, G., and Susfalk, R.: Use of heat-based vertical fluxes to approximate total flux in simple channels, Water Resources Research, 46, W03508, doi: 10.1029/2009wr007956, 2010.

Shanafield, M., McCallum, J. L., Cook, P. G., and Noorduijn, S.: Variations on thermal transport modelling of subsurface temperatures using high resolution data, Advances in Water Resources, 89, 1-9, https://doi.org/10.1016/j.advwatres.2015.12.018, 2016.

Vandenbohede, A., Louwyck, A., and Lebbe, L.: Conservative Solute Versus Heat Transport in Porous Media During Push-pull Tests, Transport in Porous Media, 76, 265-287, 10.1007/s11242-008-9246-4, 2009.

---

## Author Comment (AC2) · 19 Jan 2018

Please see attached pdf document with a detailed response to reviewer 2 comments

Please also note the supplement to this comment:
https://www.hydrol-earth-syst-sci-discuss.net/hess-2017-582/hess-2017-582-AC2-supplement.pdf
* * *

---

## Author Comment (AC3) · 19 Jan 2018

Please refer to detailed response to reviewer 2 in the pdf document

————————————————————